# Genomic and Biological Insights of Bacteriophage ΦBc24 Targeting *Bacillus cereus*

**DOI:** 10.3390/cimb47110906

**Published:** 2025-10-31

**Authors:** Nam Khang Tran, Pham Thi Lanh, Trang Trinh Thu, Man Hong Phuoc, Nguyen Dinh Duy, Vu Thi Hien, Dong Van Quyen

**Affiliations:** 1Department of Molecular Microbiology, Institute of Biology, Vietnam Academy of Science and Technology, 18 Hoang Quoc Viet, Nghia Do, Hanoi 100000, Vietnam; khangtn.d24pmab@usth.edu.vn (N.K.T.); ptlanh@ib.ac.vn (P.T.L.); trinhtrangbk1996@gmail.com (T.T.T.); phuocmh@ib.ac.vn (M.H.P.); ndduy@ib.ac.vn (N.D.D.); vuthihien@ib.ac.vn (V.T.H.); 2Department of Life Sciences, University of Science and Technology, Vietnam Academy of Science and Technology, 18 Hoang Quoc Viet, Nghia Do, Hanoi 100000, Vietnam; 3Graduate University of Science and Technology, Vietnam Academy of Science and Technology, 18 Hoang Quoc Viet, Nghia Do, Hanoi 100000, Vietnam

**Keywords:** *Bastillevirinae*, biocontrol, *Caeruleovirus*, food contamination, *Herelleviridae*, phage therapy

## Abstract

Foodborne illnesses associated with *Bacillus cereus* represent a persistent public health concern. In this study, we described the isolation and characterization of a novel bacteriophage, ΦBc24, from mud samples, which showed lytic activity against foodborne pathogen *B. cereus*. Transmission electron microscopy revealed that ΦBc24 exhibited a myovirus morphotype. Biological assays demonstrated that its narrow host range was restricted to *B. cereus* strains and efficient lytic activity, characterized by a latent period of 10 min and a burst size of 40 PFU per infected cell. The phage exhibited high physicochemical stability, tolerating pH values of 2–12, temperatures of 4–50 °C, salinity up to 1 M NaCl, and ultraviolet exposure, while effectively suppressing host bacterial growth for up to six hours. Whole-genome sequencing showed that phage ΦBc24 possessed a double-stranded DNA genome of 160,311 bp, with 39.48% GC content, and 269 predicted open reading frames (ORFs). Remarkably, 11 tRNA genes were identified, whereas no genes associated with lysogeny, virulence, or antimicrobial resistance were detected. Phylogenetic analysis suggested that ΦBc24 belongs to the genus *Caeruleovirus*, subfamily *Bastillevirinae*, family *Herelleviridae*. Taken together, these results highlight the biological robustness and genomic safety of ΦBc24, supporting its potential as a biocontrol candidate against the foodborne pathogen *B. cereus.*

## 1. Introduction

*Bacillus cereus* is an ubiquitous, Gram-positive, spore-forming bacterium with broad-spectrum pathogenicity [1,2] and is an opportunistic foodborne pathogen responsible for two types of poisoning, a diarrheal syndrome caused by enterotoxins and an emetic syndrome caused by the toxin cereulide [3,4]. For diarrheal syndrome, the organism accesses the stomach and becomes redeployed in the intestines; thus, the generation of multiple enterotoxins and other virulence factors is commenced, result in the observed symptoms, including abdominal pain, cramps, and watery diarrhea [5]. The enterotoxins correlated with diarrheal syndrome are generated in the gut and are postulated to be heat-sensitive; conversely, the toxins associated with emetic syndrome are pre-existing in food and are heat-stable [6], though they occasionally result in fatal cases of liver failure [7,8]. *B. cereus* can trigger food poisoning even at minimal doses, with cases exceeding 10^3^ cells being assessed as risky for consumption [9]. High cell counts are typically encountered in raw, non-dry ingredients and mild, heat-treated, improperly cooled products, such as raw rice [10], ready-to-eat vegetable salads, and mashed potatoes [5].

*B. cereus* has repercussions in significant foodborne occurrences worldwide, with a total prevalence of a considerable 23.746% [11]. In Europe, *B. cereus* is the second most commonly observed causal agent of identified and potential foodborne outbreaks after *Staphylococcus aureus* [12]. Conversely, this pathogen group in the USA is approximated to induce 63,400 foodborne disease cases per year, with a 0.4% hospitalization rate [13]. Furthermore, from 2010 to 2020, 419 foodborne outbreaks prompted by *B. cereus* were reported in China, leading to 7892 cases, 2786 hospital admissions, and 5 fatalities [14]. In Vietnam, *B. cereus* has also been associated with several foodborne poisoning incidents in recent years. Between 2020 and 2024, multiple outbreaks were linked to contamination by *B. cereus*, either alone or in combination with *Escherichia coli* and *Salmonella*. Notably, an outbreak in Da Nang province in 2020 affected 230 individuals, while another at a school in Nha Trang province in 2022 led to the hospitalization of more than 600 students (National Institute for Food Control, Vietnam, 2024). More recently, a food poisoning event in Phu Tho province in 2024 affected 438 workers (The National Online Conference on Strengthening Food Safety Assurance and Preventing Food Poisoning, https://vfa.gov.vn/, accessed on 22 May 2024). These incidents highlight the persistent public health risk posed by *B. cereus* contamination in food products. Despite this, *B. cereus* is not listed among the top 10 foodborne pathogens in Southeast Asia and is therefore rarely subjected to medical assessment, so its incidence is likely underestimated and should not be overlooked [15]. 

A variety of strategies have been employed to reduce food contamination in order to protect consumer health and extend product shelf life [16]. Traditionally, the food industry has relied on chemical agents, such as acids, oxidizing compounds (e.g., chlorine, H_2_O_2_), and surfactants [17,18,19]. Conventional preservation methods, including pasteurization, high-pressure processing, and UV irradiation, are also widely applied to enhance food safety [20]. However, these treatments are often associated with undesirable alterations in organoleptic properties and may contribute to the emergence of resistant pathogens [21]. Furthermore, such broad-spectrum approaches can eliminate beneficial microorganisms that are natural components of foods [22]. Antibiotics remain a primary measure for controlling pathogenic *B. cereus*, yet the rapid rise in antibiotic resistance has complicated treatment and necessitated stricter regulation of their use [9,23]. Reports have observed resistance to vancomycin, tetracycline, and erythromycin in *B. cereus*, with a survey in China showing that 13% of *Bacillus* isolates were resistant to vancomycin [24]. While vancomycin is generally considered a reliable empirical therapy against *B. cereus* infections [25], a fatal case of vancomycin-resistant *B. cereus* pneumonia and bacteremia has been documented [24]. These underscore the urgent need to develop alternative strategies for controlling pathogenic *B. cereus*.

In recent years, lytic bacteriophages (phages) have gained increasing attention as viruses that specifically infect bacterial cells and lyse their hosts, offering a promising alternative to conventional antibiotic therapy [26]. Their abundance, self-replicating capacity, and cost-effective production, as well as the absence of reported adverse effects in humans, make them highly attractive for practical applications [27]. To date, *B. cereus* phages have been studied for use as diagnostic tools, genetic engineering vectors, and promising therapeutic agents [28,29]. They can be applied either as individual phages or in phage cocktail formulations [30]. For instance, treatment of mashed potatoes with phages FWLBc1 and FWLBc2 significantly reduced *B. cereus* concentrations within 24 h, demonstrating their potential as food biocontrol agents [31]. Similarly, phages BCP1-1 and BCP8-2, isolated from Korean fermented foods, showed effective biocontrol activity against *B. cereus* in the traditional product called Cheonggukjang [32].

In this study, we report on the isolation and characterization of a novel *Bacillus* phage, ΦBc24, from mud samples collected in northern Vietnam. Its biological and genomic features were investigated to evaluate its potential as a biocontrol agent against *B. cereus* infections.

## 2. Materials and Methods

### 2.1. Bacterial Strains

Four *B. cereus* strains (VTCC 11273, VTCC 10949, VTCC 11289, VTCC 11265) were provided by the National Center for Microbial Gene Resources, Institute of Microbiology and Biotechnology (Hanoi, Vietnam). Other bacterial strains were isolated from different sources and stored in the bacterial collection in the Department of Molecular Microbiology, Institute of Biology, Vietnam Academy of Science and Technology (VAST), including *Bacillus thuringiensis*, *Bacillus subtilis* [33], *Bacillus pumilus*, *Salmonella enterica* [34], *Vibrio parahaemolyticus*, *E. coli* [35], *Staphylococcus aureus* [36] and *Lactobacillus kunkeei* [37], as shown in Appendix A.

### 2.2. Phage Isolation and Purification

Phages were isolated from mud samples collected from the To Lich River, Cau Giay, Hanoi, Vietnam (21.030063° N, 105.801074° E). Briefly, 10 g of mud sample was mixed with 100 µL of a cultured suspension of each *B. cereus* strain in 10 mL of Luria–Bertani (LB) broth (Himedia, India) and incubated at 37 °C with shaking at 120 rpm for 3 h to enrich the phages. The culture was then centrifuged at 10,000 rpm for 10 min using a Centrifuge 5415 R (Eppendorf, Germany), and the supernatant was filtered through a 0.22 µm membrane filter (Sartorius, Germany) to remove residual bacterial cells. The presence of phages was determined by plaque formation on double-layer agar plates using the agar overlay assay method [38]. This assay was performed by pouring the top agar (LB consisting of 0.7% agar) to the bottom agar (LB consisting of 2% agar), where 150 μL of bacteriophage filtrate and 150 μL of the mid-log phase bacterial host culture were mixed in 5 mL of molten LB soft agar and poured onto the LB agar plate, followed by incubation at 37 °C overnight. For phage purification, a sterile micropipette tip was inserted into the center of a plaque and gently swirled to extract a single plaque, which was then transferred into 100 μL of normal saline and subjected to a plaque assay. This purification step was repeated three times using the same host to obtain a host-specific phage. Additionally, serial dilutions of the purified phage were performed to determine phage titer (PFU/mL) [39].

### 2.3. Phage Stock Preparation

Phages were inoculated into host bacterial cultures at the mid-log phase (OD_600_ = 0.4–0.5), followed by incubation at 37 °C with shaking at 150 rpm for 3 h. After incubation, 10% (*v*/*v*) chloroform was added and mixed by vortexing, and the suspension was centrifuged at 12,000 rpm for 20 min. The supernatant was filtered through a 0.22 µm filter to remove residual bacteria. Phages were subsequently precipitated by adding 5 M NaCl (Merck, Darmstadt, Germany) and polyethylene glycol 8000 (Sigma, Burlington, MA, USA), incubated at 4 °C for 3 h, and centrifuged at 12,000 rpm for 15 min at 4 °C. The resulting pellet was resuspended in SM buffer (Himedia, Mumbai, India). Phage titers were determined using the double-layer agar method [39], and the prepared stock was used for subsequent experiments.

### 2.4. Transmission Electron Microscopy

Transmission electron microscopy (TEM) was used to determine the morphology of the phage. Phage lysate (10^9^ PFU/mL) was dropped onto carbon-coated copper grids and negatively stained with 2% uranyl acetate solution (pH 4.0) (EMS, Hatfield, PA) for 20 s. The bacteriophage’s morphology was observed using TEM JEM 1010 (Jeol, Tokyo, Japan) with a voltage of 100 kV. Phage classification was determined according to the International Committee on Taxonomy of Viruses (ICTV) [40].

### 2.5. Determination of Host Range

The host range of the isolated phage was evaluated using four *B. cereus* strains (VTCC 11273, VTCC 10949, VTCC 11289, VTCC 11265) and other bacteria, including *B. thuringiensis*, *B. subtilis*, *B. pumilus*, *S. enterica*, *V. parahaemolyticus*, *E. coli*, *S. aureus* and *L. kunkeei* (Appendix A) using the double-agar-spot assay [41]. Except for *L. kunkeei*, which was cultured in De Man–Rogosa–Sharpe (MRS) medium (Merck, Darmstadt, Germany), all other bacterial strains were grown in LB medium with shaking at 150 rpm at 37 °C. For host range determination, 150 µL of each bacterial culture (OD_600_ = 0.5–0.6) was mixed with 5 mL of molten soft agar (LB or MRS containing 0.7% agar) and overlaid onto a solid LB or MRS agar plate (2% agar). After the soft agar solidified for 15 min, 10 µL of the phage suspension was spotted onto the plate surface. The plates were incubated overnight at 37 °C, and the resulting lysis zones were examined. The phage host range was assessed based on plaque clarity: “++” indicated clear plaques; “+” indicated faint plaques; and “–“ indicated no plaque formation.

### 2.6. Multiplicity of Infection

The optimal multiplicity of infection (MOI) was determined following the previous method with some modifications [38]. *B. cereus* cultures (OD_600_ = 0.4–0.5) were mixed with phages at MOIs of 0.01, 0.1, 1, and 10, and incubated at 37 °C while shaking for 3 h. After incubation, bacterial cells were lysed with 10% chloroform, and the suspensions were centrifuged and filtered through 0.22 µm membranes. The phage titer at each MOI was determined using the double-layer agar method. All experiments were performed in triplicate, and the MOI yielding the highest phage titer was considered optimal for subsequent assays.

### 2.7. One-Step Growth Curve

The one-step growth curve of phage was determined based on the method described by Hong et al. (2014) with some modification [42]. Briefly, the phage was mixed with indicator bacteria in the mid-log phase at the optimal MOI and incubated with shaking at 37 °C for 15 min. The mixture was then centrifuged at 12,000 rpm for 1 min, and the bacterial pellet was resuspended in 20 mL of fresh LB medium. The suspension was subsequently shaken at 37 °C and 200 rpm for 90 min. Every 10 min, 100 µL sample was taken, diluted to an appropriate concentration, and the phage titer was determined using the double-layer agar method. The experiment was performed in triplicate. The latent period and the burst size were determined as described in our previous study [43].

### 2.8. Phage Stability

Phage lysates (10^8^ PFU/mL) were subjected to different stress conditions to evaluate their stability [43]. For thermal tolerance, aliquots were incubated at 4 °C, 20 °C, 30 °C, 37 °C, 50 °C, and 70 °C for an hour. For pH stability, phage suspensions were prepared in SM buffer adjusted to pH 2, 4, 6, 7, 8, 10, and 12 and incubated for an hour. Salinity tolerance was assessed by mixing phage lysates with NaCl solutions at final concentrations of 50–1000 mM (*v*/*v* = 1:1) and incubating them for an hour at room temperature. UV stability was determined by exposing the phage suspensions to UV light (254 nm, 0.6 m distance) in a biosafety cabinet (Nuaire, Plymouth, MN, USA), with 100 µL samples collected every 10 min for up to 60 min. At each time point, the samples were serially diluted, and phage titers were determined by the double-layer agar method. All experiments were performed in triplicate.

### 2.9. In Vitro Lytic Activity Assay

The lytic activity of the phage at different MOIs was evaluated following the method of Hien et al. (2025) [43], using *B. cereus* VTCC 11,273 as the host strain. The mid-log-phase culture of the host strain was mixed with the phage suspension at different MOIs (0.01, 0.1, 1, and 10). An uninfected culture of *B. cereus* VTCC 11,273 served as the control. Subsequently, the mixture was incubated at 37 °C with shaking for 12 h. An amount of 100 µL of each sample was collected every one hour, diluted and spread on the LB agar plates to enumerate the number of uninfected bacterial cells (CFU). The experiment was performed in triplicate.

### 2.10. Genome Sequencing and Annotation

Phage genomic DNA was extracted from purified particles using a modified phenolchloroform-isoamyl alcohol method as described in our previous study [43] and sequenced using the Illumina paired-end platform (2 × 150 bp). Raw reads were trimmed and quality-checked with Fastp v0.23.4 [44], and de novo assembly was performed using SPAdes v4.0.0 with multiple k-mer parameters [45]. Assembly quality was assessed with QUAST v5.2.0 [45], and the assembly graph was inspected using Bandage v0.9.0 [46]. The complete genome sequence was compared with related phages in GenBank using BLASTn (NCBI). Genome annotation was conducted with Pharokka v1.7.3 [47] to identify coding sequences (CDSs) or open reading frames (ORFs), tRNAs, mRNAs, CRISPRs, virulence factors, toxins, and antimicrobial resistance genes (ARGs). Predicted CDSs were classified into functional categories based on the PHROGs database. Genomic visualization and annotation pipelines were generated using Proksee (https://proksee.ca, accessed on 15 December 2024) [48].

### 2.11. Bioinformatic and Proteomic Analysis

Intergenomic similarities were assessed using the BLASTn algorithm implemented in VIRIDIC. Genome-wide sequence similarities were further analyzed with VipTree, based on tBLASTx comparisons, and closely related phage genomes were selected for comparative visualization [49,50]. The highly conserved protein sequences of major capsid and large terminase subunits were used for phylogenetic tree construction for the phage isolate to observe evolutionary relationships with existing GenBank database sequences. Multiple sequence alignments were performed with MUSCLE MEGA11 [51] in MEGA11, and phylogenetic trees were constructed using the maximum likelihood method.

Comparative genomic analysis was conducted using reference genomes from the INfrastructure for a PHAge REference Database (INPHARED, as of September 2024) [52]. All-vs-all BLASTp analyses and gene-sharing network classification were performed with vConTACT2 v0.9.22 [53], and networks were visualized in Cytoscape. To further resolve phylogenetic relationships, a proteomic tree was generated with VICTOR [54] using 35 reference viral genomes, including ΦBc24. Pairwise nucleotide sequence comparisons were computed using the genome-blast distance phylogeny (GBDP) method with the D6 formula [55]. Pseudo-bootstrap support values were derived from 100 replicates, and branch lengths were scaled to the respective distance formula. Taxonomic boundaries at the genus level were estimated using OPTSIL with an F value of 0.5 [55,56].

### 2.12. Statistical Analysis

Statistical analyses were performed using RStudio version 2024.12.0+467. Descriptive statistics were applied to calculate means, standard deviations, and percentages, as well as to generate tables and graphical representations of phage lysis profiles and stability. Differences in phage titers among experimental groups were evaluated by one way-ANOVA, followed by a Tukey test to see if any group differed significantly further from the others. The results were considered statistically significant at *p* < 0.05.

## 3. Results

### 3.1. Phage Isolation

In this study, a *B. cereus* phage, designated ΦBc24, was successfully isolated using the double-agar method. ΦBc24 exhibited potent lytic activity against all four tested *B. cereus* strains (Appendix A). Among the susceptible strains, the highest lytic activity was observed against *B. cereus* VTCC 11273, which was subsequently used as the propagation host for phage enrichment and titer determination in subsequent experiments. When propagated on *B. cereus* VTCC 11273, the phage reached a titer of approximately 2.5 × 10^9^ PFU/mL. The plaques produced by phage ΦBc24 was shown in Figure 1a.

### 3.2. Phage Morphology

Transmission electron microscopy (TEM) revealed that phage ΦBc24 possessed a head–tail structure, as shown in Figure 1b, with an icosahedral head measuring 93.3 ± 2.5 nm and a contractile tail measuring 131 ± 3 nm when contracted and 174 ± 4 nm when extended. Based on ICTV (2022) classification criteria, ΦBc24 was identified as a myovirus-like phage [40].

### 3.3. Host Range

Phage ΦBc24 exhibited narrow and strictly host-specific lytic activity, lysing only tested *B. cereus* strains. In contrast, no lytic activity was observed against any other *Bacillus* species (*B. thuringiensis*, *B. subtilis* and *B. pumilus*) or against non-*Bacillus* species (*S. enterica*, *V. parahaemolyticus*, *E. coli*, *S. aureus* and *L. kunkeei*) tested, as shown in Appendix A. This host specificity highlights the high degree of phage–host adaptation characteristic of the phage ΦBc24.

### 3.4. Optimal Multiplicity of Infection

The multiplicity of infection (MOI), defined as the ratio of phage particles to host cells, is a critical parameter for maximizing phage production. As shown in Figure 2a, phage ΦBc24 yielded high titers across all tested MOIs (0.01, 0.1, 1, and 10). The highest titer, approximately 9.2 log PFU/mL, was obtained at an MOI of 0.01, followed by phage titers at MOIs 0.1 and 1 of about 9 log PFU/mL. The lowest value was recorded at an MOI of 10 (8.36 log PFU/mL). Accordingly, an MOI of 0.01 was determined to be optimal for phage ΦBc24 and was employed in subsequent experiments.

### 3.5. One-Step Growth Curve

Phage ΦBc24 exhibited a short latent period of approximately 10 min and two distinct latent phases after 90 min, as shown in Figure 2b. A sharp increase in phage titer was observed at 10 min post-infection, followed by a second rise at 40 min. The growth curve reached a stationary phase after 60 min, with a maximum titer of approximately 9.2 log PFU/mL, and the average burst size was estimated at 40 PFU per infected cell.

### 3.6. Phage Stability

The stability of phage ΦBc24 was evaluated under various conditions of temperature, pH, salinity, and UV exposure. As a result, phage ΦBc24 was heat-stable at temperatures below 50 °C, with average titers maintaining unchanged (*p* > 0.05), but its titer significantly declined to 2.6 log PFU/mL (*p* < 0.05) after 1 h at 70 °C (Figure 3a). Similarly, the lytic activity of phage ΦBc24 remained stable across a wide pH range (pH 2–12) (*p* > 0.05) (Figure 3b). Moreover, the phage also demonstrated considerable tolerance to UV irradiation, with only a reduction of about 2.1 log PFU/mL (*p* < 0.05) after 60 min of exposure (Figure 3c). In addition, NaCl concentrations up to 1 M did not significantly affect ΦBc24 titers as shown in Figure 3d.

### 3.7. In Vitro Lytic Activity of Phage ΦBC24

Phage ΦBc24 markedly inhibited the growth of the tested bacterium and delayed its transition to the stationary phase relative to the control group, as shown in Figure 4. In the control group, the bacterial cell density gradually increased during the experimental period, reaching the stationary phase after 8 h of incubation. In contrast, all ΦBC24-treated cultures exhibited marked growth inhibition during the first 3 h. The bacterial concentration, however, subsequently increased by about 1.8–2.0 log CFU/mL, but was significantly suppressed over the following 3 h. Although partial regrowth was observed thereafter, the stationary phase was reached later, at 11 h, and at a lower cell density (~7.3–7.8 log CFU/mL compared with the control (~10 log CFU/mL).

### 3.8. Characterization of Phage ΦBc24 Genome

The phage ΦBc24 had a circular double-stranded DNA genome of 160,311 base pairs, with a GC content of 39.48%. The complete genome sequence of this phage was deposited in the NCBI database under BioProject PRJNA1311713 and GenBank accession number PX259839. The ΦBc24 genome contains 269 ORFs with 92 ORFs annotated and 177 ORFs assigned as hypothetical proteins using PHROG databases. The genome is observed to contain 11 tRNA genes, and all ORFs were categorized into different cluster functional groups, including DNA replication and metabolism, head and packaging, auxiliary metabolic gene, lysis, hypothetical proteins and others, as represented in Figure 5 and the Appendix A.

Among the 92 ORFs annotated, ORF_221 and ORF_220 encoded terminase small subunits and terminase large subunits, respectively, while ORF_240 and ORF_241 encoded portal proteins clustered together for DNA packaging groups. Furthermore, the structural module consists of a total of 30 annotated genes involved in structural proteins and virion assembly, including 12 ORFs that are classified in the head protein group, as well as two putative ORFs, which are ORF_254 (L-alamyl-D-glutamate peptidase) and ORF_260 (glycerophosphoryl diester phosphodiesterase). The individuals responsible for forming the capsid were ORF_5, 9, 92, 247, 252 and 267 (virion structural protein); ORF_244 (major head protein); ORF_242 (head maturation protease). The gene immediately downstream of the group encodes a set of tail clustering proteins which are expressed by a series of genes, including ORF_7 and 269 (baseplate protein); ORF_3, 237, 246 and 264 (tail fiber protein); ORF_261, 262, and 263 (tail protein); ORF_259 (tail with lysin activity); ORF_258 (tail-associated lysin); ORF_256 and 257 (tail assembly chaperone); ORF_251 (tail sheath); ORF_8 (baseplate wedge subunit). The lytic module cluster group was upstream of the structure groups of the ΦBc24 genome. Two ORFs, ORF_218 (endolysin) and ORF_49 (holin), are assigned, forming a classic holin-endolysin lysis system. Moreover, 27 genes involved DNA, RNA, and nucleotide metabolism. For instance, ORF_14 encodes DNA helicase, while in the same cluster, ORF_15, 36, 72, and 93 are assigned for DNA binding protein; ORF_41 encodes DNA polymerase, ORF_21 encodes DNA primase, and ORF_23 is annotated as dUTPase; four ORFs (ORF_114, ORF_208, ORF_231, and ORF_235) were defined as HNH endonuclease (Appendix A).

The genome sequence of phage ΦBc24 was compared with other phage genome sequences available in the NCBI database using BLASTn. As a result, the genomic sequence of ΦBc24 exhibited the highest similarity to other *Bacillus* phages isolated from sewage and soil samples, including CM1 (GenBank accession no. 3093841, Iran), BM15 (GenBank accession no. 1755680, Egypt), and PK16 (GenBank accession no. KX495186.1, Korea), as presented in Figure 6, all of which were classified within the *Herelleviridae* family. Moreover, a VIRDIC heatmap further demonstrated that phage ΦBc24 shared the highest nucleotide sequence identity with CM1 (81.4%), BM15 (77.7%), and PK16 (76.4%), as shown in Figure 7, indicating that these phages belong to the same genus. Therefore, phage ΦBc24 is likely a member of the genus *Caeruleovirus*, consistent with the classification of these related phages.

### 3.9. Phylogenetic Analyses of Phage ΦBc24

Phylogenetic analyses based on the major capsid protein and the terminase large subunit revealed that phage ΦBc24 clustered with other *Bacillus* phage members of the family *Herelleviridae*, indicating a close evolutionary relationship (Figure 8a,b). Comparative genomic analysis using vConTACT2 further revealed that ΦBc24 formed a distinct sub-cluster (VC_0_1) with 27 other *Bacillus* phages, all classified within the subfamily *Bastillevirinae*, as shown in Figure 9. Within this sub-cluster, two novel genera, *Tsarbombavirus* and *Caeruleovirus*, were identified together with phage ΦBc24. Separately, genome-based phylogenetic analysis using the GBDP method demonstrated that the D6 formula tree resolved two genus-level and two subfamily-level clusters within one single family cluster, with an average support of 46% (Figure 10). In this phylogenetic tree, phages in cluster VC_0_1 were separated into two distinct branches, where ΦBc24 aligns with reference phages assigned to the genus *Caeruleovirus*, thereby reinforcing the taxonomic placement suggested by previous analyses from VIRIDIC.

## 4. Discussion

Food safety remains a pressing global challenge, exacerbated by the rise in antimicrobial-resistant bacteria [10,57]. Phages are natural antibacterial agents with strong biocontrol potential against major and emerging foodborne pathogens such as *Bacillus*, *Campylobacter*, *E. coli*, *Listeria monocytogenes*, *Salmonella*, *Shigella*, and *Vibrio* spp. [58,59,60,61]. During the lytic cycle, each phage produces numerous progeny that rapidly lyse host cells, effectively suppressing pathogens [62]. Compared with traditional antimicrobials, phages are highly specific, preserve beneficial microbiota, have no adverse effects on humans, do not alter food quality, are easy and low-cost to produce, are stable under diverse conditions, and are self-replicating—eliminating repeated dosing [63]. These attributes make them valuable for both the detection and control of pathogens throughout the food chain, including post-harvest surface decontamination, antimicrobial packaging, equipment sanitation, animal pre-harvest treatment, and phage-based diagnostics [59,63,64]. Here, we report on the isolation and characterization of a novel lytic phage, ΦBc24, from mud samples collected from the To Lich River in Cau Giay, Hanoi, Vietnam, and evaluate its potential as a biocontrol agent against *B. cereus*.

Based on TEM image analysis and ICTV guidelines [40], phage ΦBc24 exhibits a myovirus morphology, characterized by an icosahedral head measuring 93.3 ± 2.5 nm and a long contractile tail measuring 131 ± 3 nm when contracted and 174 ± 4 nm when extended. Such structural features are typical of members of the class *Caudoviricetes* (formerly order *Caudovirales*), which comprise approximately 96% of all known phages and are considered the most abundant biological entities on Earth [65]. Some *Bacillus* phages within the class *Caudoviricetes* have recently been reported, such as phage DZ1 (*Andromedavirus* genus, *Ehrlichviridae* family) [4], phage DLn1 (*Northropvirinae* subfamily, *Salasmaviridae* family) [66], phage LysPBC4 [67], phage Deep-Purple [68], phage vB_BceP_LY3 (subfamily *Northropvirinae*) [69], phage vB_BceM-HSE3 [70], phage DC1 and DC2 (*Herelleviridae* family) [71].

Previous studies have demonstrated that host specificity is a critical determinant in the application of phages [72]. In this study, phage ΦBc24 exhibited strict host-specific lytic activity, infecting only *B. cereus* strains without affecting other tested pathogenic or beneficial bacteria, thereby minimizing collateral impacts on environmental microbiota [26]. The phage had an optimal MOI of 0.01, a short latent period of 10 min, and a burst size of 40 PFU per infected cell. In comparison, phages DC1 and DC2, isolated from water samples in Guangzhou, China, exhibited higher optimal MOIs of 10 and 1, respectively. DC1 had a latent period of 30 min and a burst size of 39 PFU/cell, whereas DC2 displayed a latent period of 15 min and a burst size of 124 PFU/cell [71]. Similarly, phage SWEP1 displayed a 20 min latency with a burst size of 83 PFU/cell [73]. In addition, the in vitro assay revealed that ΦBc24 significantly inhibited *B. cereus* growth for up to six hours and delayed its entry into the stationary phase, underscoring its potential as a promising biocontrol candidate against *B. cereus* infections.

For effective application, bacteriophages must remain stable under diverse environmental conditions. Temperature is a key factor influencing phage adsorption, genome injection, replication, and latency [74], whereas pH strongly affects stability and infectivity, with acidic environments often causing aggregation or inactivation [75,76]. In this study, we found that phage ΦBc24 demonstrated broad thermal stability, remaining active between 4 °C and 50 °C, and exhibited remarkable pH tolerance, retaining infectivity across a wide pH range (2–12). By comparison, phage SWEP1 (*Herelleviridae*) was stable below 50 °C but inactivated at 60 °C, with activity limited to pH 4–11 [73]. Similarly, phages DC1 and DC2 displayed narrower stability ranges (4–45 °C; pH 4–9) [71]. In addition, ΦBc24 tolerated salinity up to 1 M and exhibited moderate resistance to UV irradiation (λ = 254 nm). The broad physicochemical stability of ΦBc24 underscores its robust adaptation to variable environments, which could refer to the phage capsid and long contractile tail characteristic of the phage [77,78] and highlights its potential as a biocontrol agent against *B. cereus* contamination. The lytic activity of ΦBc24 was evaluated over 12 h to determine its short-term infection dynamics against *B. cereus*. Because *Bacillus* species can form spores that are resistant to phage infection, longer-term studies including 48–96 h growth monitoring and spore revival assays will be conducted in future work to fully assess the sustainability of ΦBc24’s antibacterial effect and its potential for food preservation applications.

Genomically, phage ΦBc24 had a circular double-stranded DNA genome of 160,311 bp with a GC content of 39.48%. This genome size is comparable to several previously reported *Bacillus* phages, including DC1 (156,018 bp) and DC2 (155,908 bp) [71], Thurquoise (157,500 bp) [79], and vB_BceM-HSE3 (124,002 bp) [70]. In contrast, some *Bacillus* phages carry considerably smaller genomes, such as Deep-Purple (Siphovirus, 36,278 bp) [68], PBC1 (41.2 kb) [80], or vB_BceP_LY3 (subfamily *Northropvirinae*) (28,124 bp) [69].

In line with other *B. cereus* phages, the genome of ΦBc24 encodes the core functions required for host infection, intracellular replication, phage protein synthesis, and progeny assembly. Beyond these essential elements, ΦBc24 harbors genes associated with morons, auxiliary metabolic functions, host takeover, phage adaptation, and resistance to environmental stresses, reflecting an enhanced genetic capacity compared to many previously characterized *B. cereus* phages. Such features suggest that ΦBc24 is evolutionarily optimized to persist and remain active under diverse ecological conditions, thereby reinforcing its potential as a robust biocontrol agent. Bacteriophages, as viruses that exclusively infect bacteria, must locally degrade host cell barriers to initiate infection, a process mediated by lytic enzymes (lysins) located primarily in the phage tail. These enzymes cleave specific bonds within the peptidoglycan layer of the bacterial cell wall, resulting in hypotonic lysis and the release of progeny virions [81,82]. It is well established that holins, endolysins, and spanins constitute the key components driving bacterial cell lysis [83,84]. Holins, which are small membrane proteins, form aggregates in the cytoplasmic membrane to generate pores, thereby acting as molecular “timers” that regulate the onset of lysis [85]. Once these pores are established, endolysins, powerful peptidoglycan hydrolases, gain access to the bacterial cell wall, degrade peptidoglycan, and complete host cell lysis [84]. When these lytic proteins reach a critical concentration, the infected cell undergoes lysis, leading to the extensive release of progeny virions, a phase commonly referred to as the lysis period [86]. In the present study, both endolysin and holin were identified in the ΦBc24 genome, forming a canonical holin and endolysin lysis system strategically located upstream of the structural gene cluster. This genetic organization underscores the phage’s optimization for rapid and efficient host cell lysis, conferring ΦBc24 a distinct advantage in terms of lytic potency, host suppression, and applicability in controlling *B. cereus* and related pathogens.

Moreover, our data indicated that the genome of phage ΦBc24 also encodes multiple enzymes involved in host cell wall degradation during the lytic cycle, including metal-dependent hydrolases, tail-associated lysins, tail proteins with lysin activity, and L-Ala-D-Glu peptidase. The latter, also known as L-alanyl-D-glutamate endopeptidase, cleaves the L-alanine–D-glutamic acid linkage within peptidoglycan, leading to host cell lysis and the release of progeny virions [87,88]. Because this bond is universally present in type A peptidoglycan, characteristic of all Gram-negative and most Gram-positive bacteria, such enzymes are considered broad-spectrum antimicrobials (“enzybiotics”) with potential therapeutic and biotechnological applications, including the treatment of antibiotic-resistant infections [88].

In addition, spore-forming bacteria such as *Bacillus* spp. can evade phage infection through sporulation; however, phages may counter this defense by expressing host-like or phage-encoded sporulation genes, reflecting coevolutionary dynamics [89,90]. In the present study, the phage ΦBc24 genome was found to contain several such genes, including an FtsK/SpoIIIE-like protein [89,91,92], a sporulation protein [89], an anti-sigma factor [93], an ocr-like anti-restriction [94], and a UvsX-like recombinase [95], suggesting its potential strategies to overcome host defenses.

Phage-encoded tRNAs have been proposed to facilitate integration into host chromosomes, optimize codon usage, and enhance translational efficiency, particularly during late-stage infection when host rRNA pools may be degraded. In addition, they may prolong the replication period during host cell shutdown and mitigate selection pressure during long-term infections [49]. However, the number of tRNA in phage genomes varies. For instance, DC1 and DC2 [71], and Thurquoise were found to encode tRNAs [79], whereas other phages, such as Z3 (subfamily *Bastillevirinae*, family *Herelleviridae*), were found to not contain any tRNAs [96]. In this study, the genome of phage ΦBc24 was found to encode 11 tRNA genes, which may enhance translational efficiency, reduce dependence on host translational machinery, and promote replication under diverse host conditions. Importantly, no genes related to antibiotic resistance, virulence, or lysogeny were detected, underscoring the potential safety of phage ΦBc24 for biocontrol applications. Although the current genomic analyses support the genetic safety of this phage, we acknowledge the limitations of sequence-based inference, and additional biosafety evaluations via in vivo assessments are required to confirm this finding.

According to ICTV guidelines, phage taxonomy should be based on genomic and proteomic relatedness, with novel isolates assigned to an existing genus if they share more than 70% nucleotide identity or if more than 40% of their proteins exhibit at least 70% amino acid identity with members of that genus [97]. A series of analyses were carried out to fulfill ICTV requirements. In the present study, comparative analysis of the phage ΦBc24 genome against other phage sequences in the NCBI database was performed using BLASTn and VIRIDIC. The results showed the high identity of phage ΦBc24 genome with other *Bacillus* phages, including CM1, BM15, and PK16, all of which are classified within the genus *Caeruleovirus* of the family *Herelleviridae* [98]. Furthermore, to explore the evolutionary relationships between ΦBc24 and other tailed phages and to strengthen the prediction of the taxonomy position of ΦBc24, phylogenetic trees were generated for the major capsid proteins and the terminase large subunit, two proteins often used as markers in phage phylogenetic analysis [83]. The large terminase subunit encodes an ATP-driven motor responsible for genome packaging into the viral capsid and is highly conserved among *Caudoviricetes* phages, making it a reliable phylogenetic marker [99,100,101]. Meanwhile, the major capsid protein serves as the gold standard for lytic phage taxonomy [102,103]. The major capsid protein gene encodes the principal structural protein forming the outer capsid that encapsulates and protects the viral genome during infection [102,104]. Due to its essential structural role and high conservation within specific phage families, the major capsid protein is frequently used to construct phylogenetic trees and infer evolutionary relationships among phage isolates [102]. In the present study, ΦBc24 formed a branch with other *Herelleviridae* phages, indicating close relatives among sequenced bacterial viruses. Comparative genomic network analysis using vConTACT2 confirmed its placement within the same genus, strengthening the assignment of ΦBc24 to *Caeruleovirus*.

Furthermore, a genome-based network analysis of phage ΦBc24 was performed using vConTACT2 with reference databases from INPHARED [53]. As a result, phage ΦBc24 was placed within sub-cluster VC_0_1 of the *Herelleviridae* family, together with 27 other *Bacillus* phages classified under the subfamily *Bastillevirinae*, which comprises large myoviruses infecting *Bacillus* species [79]. Within sub-cluster VC_0_1 of the *Bastillevirinae* subfamily, phage ΦBc24 was grouped alongside two recently defined genera, *Tsarbombavirus* and *Caeruleovirus*. To further resolve its taxonomic placement, genome-based phylogeny was conducted using the GBDP method [55,56], incorporating all 27 *Bacillus* phages from the same sub-cluster together with seven phages from other *Herelleviridae* subfamilies, consistent with ICTV network-based classification criteria [54]. The D6 formula GBDP tree yields two genus clusters and two sub-family clusters within one single family cluster, with an average support of 46%. While ICTV taxonomy distinguished two genera within the same Vcontact2 sub-cluster, the BLAST-based VICTOR phylogenomic framework indicated that all phages in VC_0_1 belonged to a single genus, separating only at the subfamily level. Within VC_0_1, ΦBc24 clustered with reference phages of the genus *Caeruleovirus*. Consistent with VIRDIC, both genome-based network analysis and whole-genome proteomic phylogeny confirmed that ΦBc24 belongs to the genus *Caeruleovirus*, subfamily *Bastillevirinae*, and family *Herelleviridae*. These findings not only clarify the evolutionary position of phage ΦBc24 but also provide a framework for identifying related phages with potential biotechnological applications.

In recent years, phages targeting *B. cereus* have garnered increasing attention for their potential to reduce *B. cereus* contamination in food production across several countries, including Indonesia (2024) [105], China (2024) [4,71], Korea (2017 and 2021) [21,106], and Belgium (2014) [107]. However, large-scale implementation of phage-based strategies remains in its early stages. This study demonstrated phage ΦBc24 as a promising candidate for innovative, non-invasive strategies to control foodborne pathogen *B. cereus*. Our results provided additional evidence, resources, and foundational knowledge regarding *B. cereus* phages and their potential applications in food production since the number of available *B. cereus*-infecting phage isolates are relatively small and products specifically targeting *B.cereus* remain limited [27].

As a natural, highly specific, and environmentally sustainable agent, ΦBc24 offers advantages in enhancing food safety, extending shelf life, and reducing reliance on antibiotics and chemical preservatives, with potential applications from primary production to biopreservation. However, the inherently narrow host range of individual phages, including ΦBc24, underscores the necessity of employing phage cocktails with broader and polyvalent host specificity to achieve more comprehensive antibacterial efficacy and to mitigate the risk of resistance development.

In addition to the phage itself, phage-encoded lytic enzymes, particularly endolysins, offer valuable alternatives for combating pathogenic bacteria [82,108]. For instance, PlyHSE3, an endolysin from phage vB_BceM-HSE3, displayed a broader lytic spectrum than its parental phage, effectively lysing all tested *B. cereus* group strains as well as *P. aeruginosa* [70]. Likewise, PlyB has been shown to be a promising therapeutic agent against *B. cereus* eye infections [82]. Given that the ΦBc24 genome encodes multiple lytic enzymes, both the phage and its lytic proteins represent promising candidates for controlling *B. cereus*, particularly antimicrobial-resistant strains. This is particularly significant in Vietnam, where *B. cereus* is a prevalent cause of foodborne illness and numerous isolates from diverse sources exhibit multidrug resistance [109,110], thereby posing a substantial public health concern. The distinctive genetic and functional features of ΦBc24 emphasize its potent lytic activity, stability, and adaptability, supporting its potential as both a biocontrol agent and a source of therapeutic enzymes. Overall, phage-based strategies utilizing ΦBc24 represent a potential candidate for the development of safer and more sustainable approaches to control *B. cereus*, ultimately contributing to improved food safety and public health. Nonetheless, additional studies are warranted to assess the stability and antibacterial activity of ΦBc24 in model food systems, which will help determine its practical feasibility as a biocontrol agent. Furthermore, successful commercialization will depend on bridging current scientific and technological gaps.

In fact, phages are increasingly recognized as safe (GRAS) for food use and considered organic and legitimate [61,111,112]. Since the first FDA-approved phage product (ListShield™, 2006), multiple preparations—including SalmoFresh™ (2013), ShigaShield™ (2017), and Salmonelex™ (2013)—have entered the market from companies such as Intralytix, Micreos, FINK TEC, Passport Food Safety Solutions, and Phagelux [58,62]. These developments highlight growing confidence in phage efficacy and safety.

However, widespread commercialization is constrained by factors such as narrow host range, inconsistent database annotations and numerous uncharacterized ORFs, complicating safety interpretation, formulation stability, limited real-world validation (in large-scale and multi-site trials) and evaluation standards [62,63,111,112]. Moreover, legal constraints and regulatory frameworks vary internationally—while GRAS/FCN approvals exist in the U.S., phages are identified as novel innovations in the EU and require EFSA evaluation. Canada, Australia, Switzerland, Israel, and New Zealand have also approved phage products, often aligning with U.S. precedents. However, slow EU regulatory adoption continues to delay implementation. Therefore, harmonized evaluation standards and transparent risk–benefit communication remain essential for global deployment [112].

To ensure safe and responsible commercialization, standardized genomic screening pipelines, harmonized annotation practices, and large-scale and multi-site field trials are needed to confirm robustness and monitor resistance. Improved formulation technologies can enhance stability and delivery, while early regulatory engagement may streamline approval. Lastly, transparent, evidence-based consumer communication is vital to build public understanding and trust in phage-based solutions [59,63,112]. Collectively, these efforts will support the global transition of phage-based control strategies from laboratory research toward practical and widespread application.

## 5. Conclusions

This study reports, for the first time, the isolation and genomic characterization of *B. cereus* phage ΦBc24 in Vietnam. Phage ΦBc24 represents a novel lytic member of the family *Herelleviridae*, subfamily *Bastillevirinae*, and genus *Caeruleovirus*. Detailed biological and genomic analyses demonstrated its pronounced host specificity, potent lytic activity, and absence of undesirable genetic elements, underscoring its potential as a safe and effective biocontrol agent against *B. cereus*. Collectively, these findings reinforce the promise of bacteriophage-based biocontrol as a sustainable and targeted alternative to conventional antimicrobial approaches for mitigating foodborne pathogens and enhancing food safety. Further investigations are warranted to optimize formulation and delivery strategies, evaluate long-term stability and safety, and address potential implications for human health and regulatory acceptance.

## Figures and Tables

**Figure 1 cimb-47-00906-f001:**
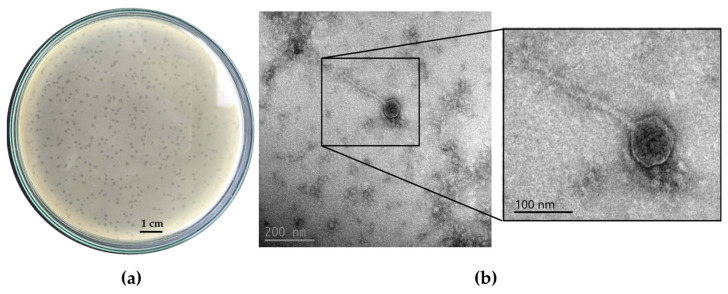
Morphology of phage ΦBc24. (**a**) Plaque morphology of ΦBc24 after incubation with *B. cereus* VTCC 11,273 on a double-layer agar plate after overnight culture, (**b**) virion morphology of ΦBc24 by transmission electron microscopy (TEM) with an icosahedral head measuring 93.3 ± 2.5 nm and a contractile tail measuring 131 ± 3 nm when contracted and 174 ± 4 nm when extended.

**Figure 2 cimb-47-00906-f002:**
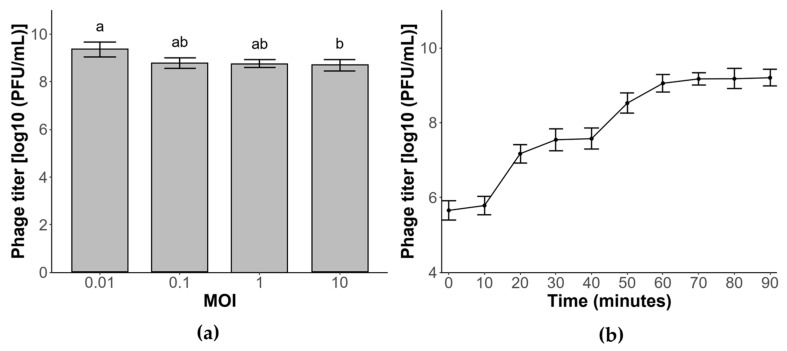
Biological properties of phage ΦBc24 against *B. cereus* VTCC 11273. (**a**) Multiplicity of infection; (**b**) one-step growth curve. All values are reported as averages from triplicate experiments. The error bars represent the 95% confidence interval of the data set. Statistically significant differences were determined by a Tukey test. Different letters (a,b) indicate statistically significant differences (*p* < 0.05).

**Figure 3 cimb-47-00906-f003:**
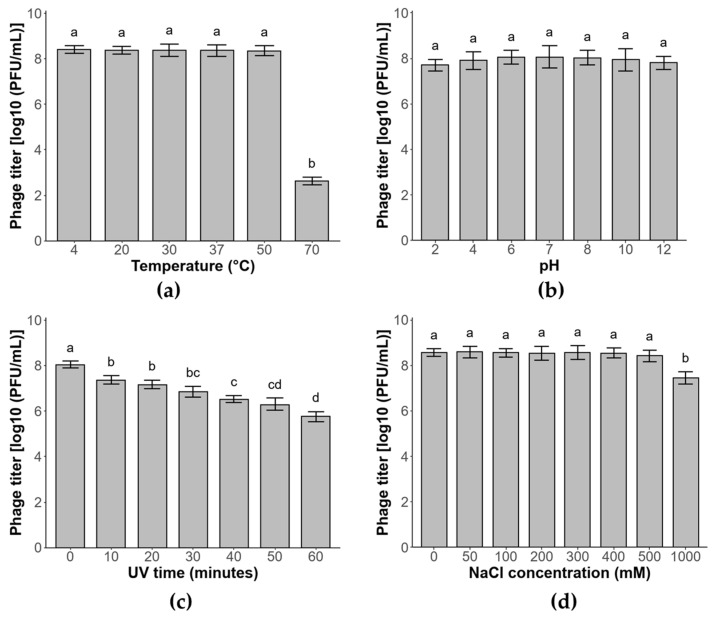
Stability of phage ΦBc24 under different conditions. (**a**) Phage viability at different temperatures, (**b**) phage viability at different pH levels, (**c**) phage viability under UV exposure and (**d**) phage viability under different salinity concentrations. All values are reported as averages from triplicate experiments. The error bars represent the 95% confidence interval of the data set. Statistically significant differences were determined by a Tukey test. Different letters (a–d) indicate statistically significant differences (*p* < 0.05).

**Figure 4 cimb-47-00906-f004:**
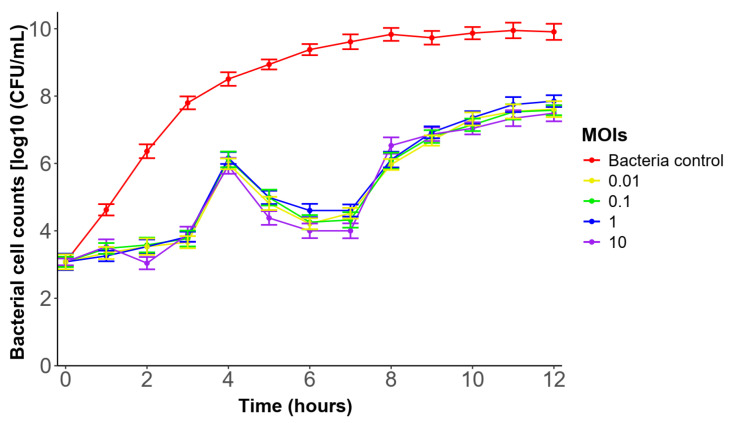
In vitro lytic activity of phage ΦBc24 at various multiplicities of infection (MOIs) against *B. cereus* VTCC 11273. Four different MOIs (0.01, 0.1, 1, and 10) were used for infecting *B. cereus* VTCC 11273. The red line represents a standard growth curve of *B. cereus* VTCC 11,273 in solution as the bacterial control. All values are reported as averages from triplicate experiments. The error bars represent the 95% confidence interval of the data set.

**Figure 5 cimb-47-00906-f005:**
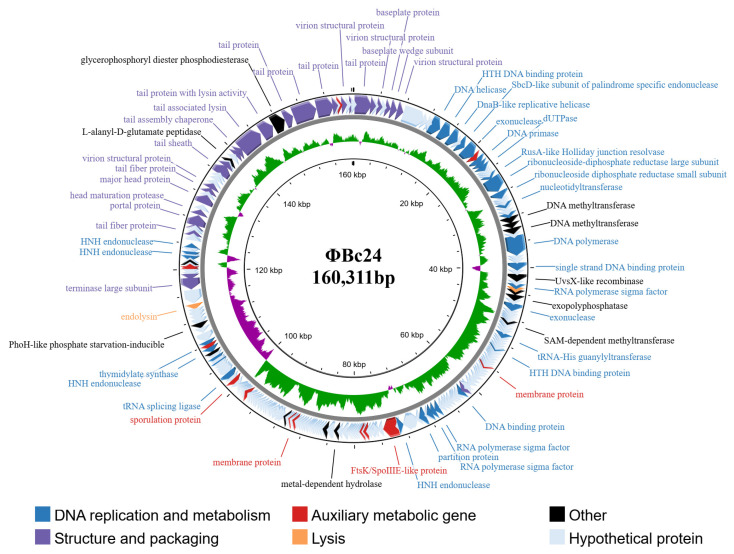
Genome map of phage ΦBc24. From the innermost to the outermost circles, the circle represents GC content (black), GC skew (green indicates GC skew+ and purple indicates GC skew−), and predicted ORFs (clockwise for the forward strand and counterclockwise for the reverse strand).

**Figure 6 cimb-47-00906-f006:**
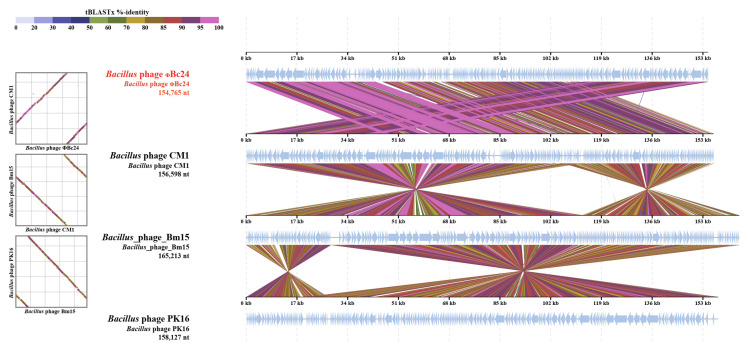
Genome comparison of phage ΦBc24 (highlighted in red) and closely related phages using BLASTn.

**Figure 7 cimb-47-00906-f007:**
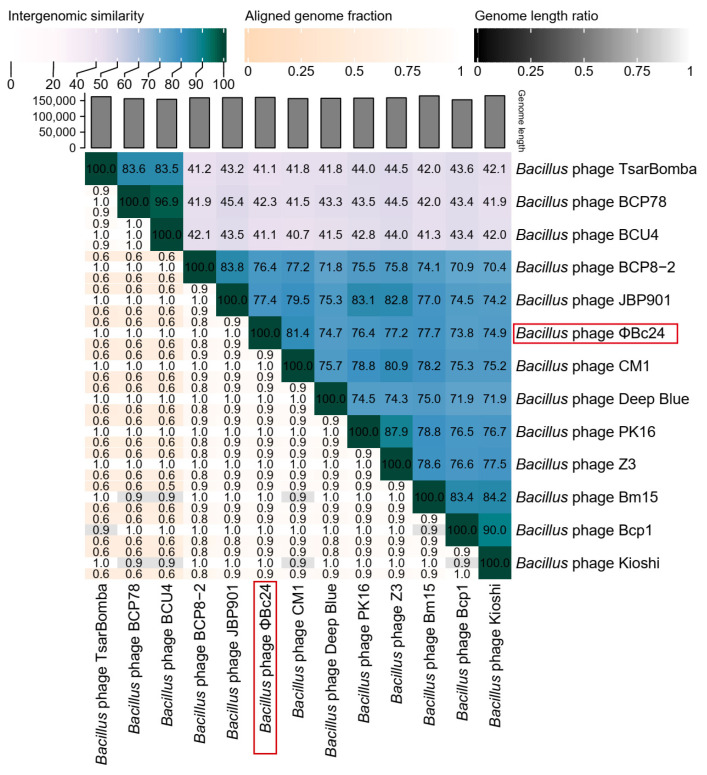
VIRIDIC heatmap based on intergenomic similarities amongst viral genomes, with the color scale indicating similarity percentages. Phage ΦBc24 is highlighted with a red box.

**Figure 8 cimb-47-00906-f008:**
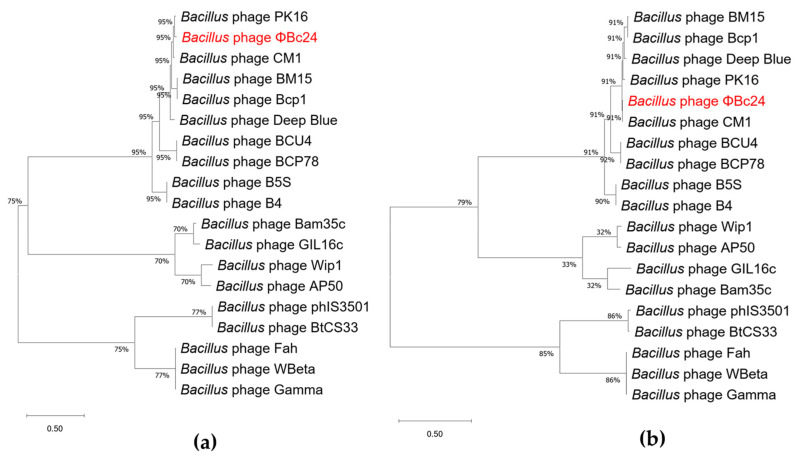
Maximum likelihood phylogenetic trees of phage ΦBc24 (highlighted in red) and other *Bacillus* phages. (**a**) Phylogenetic tree based on the major capsid protein; (**b**) phylogenetic tree based on the terminase large subunit.

**Figure 9 cimb-47-00906-f009:**
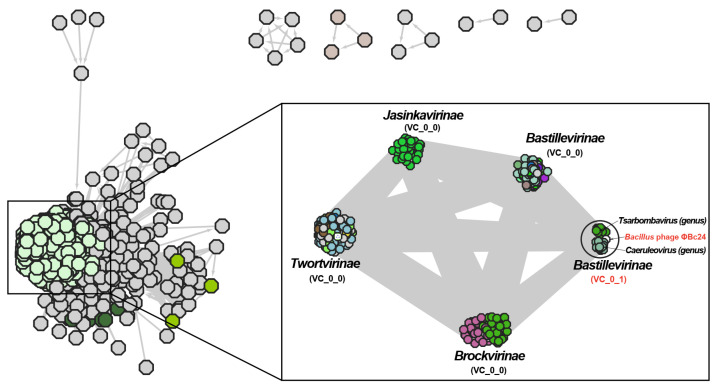
Comparative genomic analysis of phage ΦBc24 with other *Bacillus* phages using vConTACT2 with reference databases from INPHARED. Phage ΦBc24 and sub-cluster (VC_0_1) are highlighted in red.

**Figure 10 cimb-47-00906-f010:**
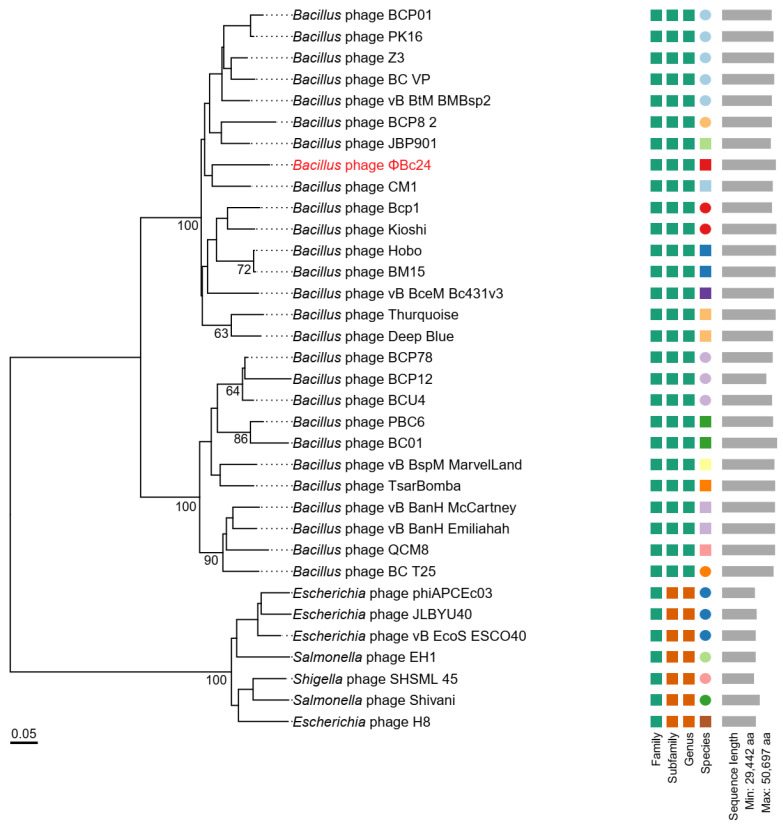
Phylogenetic tree based on Genome-BLAST Distance Phylogeny (GBDP) of phage ΦBc24 (highlighted in red) and other phages from the ICTV database.

## Data Availability

The genome sequence of phage ΦBc24 was deposited in the NCBI database under the BioProject PRJNA1311713, GenBank accession number PX259839.

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
