# Peer review of "Genomic and Biological Insights of Bacteriophage ΦBc24 Targeting Bacillus cereus"

_cimb, 2025, doi:10.3390/cimb47110906_

Round 1

Reviewer 1 Report

Comments and Suggestions for Authors

Bacillus cereus, a foodborne pathogen known for causing diarrhea and vomiting, is increasingly resistant to drugs. This study identified a novel bacteriophage, Φ Bc24, in mud from Hanoi, Vietnam. Φ Bc24 exhibits specific targeting of Bacillus cereus, strong environmental stability, and safe genomic characteristics, indicating its significant potential as a biological control agent for food contamination.

The experimental design is reasonable, and the paper is logically structured. Supported by relevant data, the presentation is clear, and most conclusions are well-justified. However, there are some concerns regarding the experiment that need to be addressed before acceptance.

  1. Please check the whole manuscript carefully for avoiding grammar errors or spelling mistakes.
  2. Please introduce the full name in the initial mention and link it with the corresponding acronym.(g. Line 23 ORFS, Line 125 PEG 800)
  3. Materials and Methods: Please specify the primary instruments and reagents used, including their specific models and manufacturers.(e.g. LB, Centrifuge)
  4. Line 90 to 97 The study employed four standard B. cereus strains, including VTCC 11273, and seven non-target bacteria, such as B. subtilis and E. coli. The test strains comprised species of the genus Bacillus and a few Gram-negative bacteria (Supplementary Table 1). Excluding common food industry contaminants, like close relatives of B. thuringiensis or B. anthracis, might result in an overestimation of host specificity.
  5. Line 162 to 171 The physicochemical stability experiments were conducted in SM buffer (pH, temperature, UV, etc.) without using food matrices. Buffers cannot simulate the effects of complex food components (such as fats, proteins) on phage activity. There is a lack of direct evidence to extrapolate the conclusions to 'food biocontrol agent' (Abstract, Line 27).
  6. Line 178 In vitro lytic activity assay experiment was observed for only 12 hours, and neither spore revival nor long-term antibacterial effects were evaluated, making it challenging to support its application in real food preservation scenarios.(Introduction, Line 44)
  7. Figure 1 The scale lacks clarity. The enlarged image should include a scale to facilitate easier observation and comparison.
  8. Line 199 Why choose protein sequences of major capsid and large terminase subunits for phylogenetic analysis? Instead of analyzing the evolutionary branches of genes directly associated with lysis efficiency such as lysozyme and perforin. This may limit the understanding of the unique lysis mechanism of ΦBc24 and reduce its differential competitiveness compared to similar bacteriophages.
  9. Despite bioinformatics predicting 'non-virulence,resistance genes,' experimental validation, such as hemolysis tests or detection of resistance gene expression, was not conducted. Relying solely on sequence prediction could overlook unannotated novel virulence factors, potentially posing biological safety risks, particularly when used as a food additive.

Author Response

Genomic and Biological Insights of Bacteriophage ΦBc24 Targeting Bacillus cereus

Response to Reviewer 1 Comments

1. Summary

Thank you very much for taking the time to review this manuscript. Please find the detailed responses below and the corresponding revisions in track changes in the re-submitted files.

2. Questions for General Evaluation

Reviewer’s Evaluation

Does the introduction provide sufficient background and include all relevant references?

Can be improved

Is the research design appropriate?

Can be improved

Are the methods adequately described?

Can be improved

Are the results clearly presented?

Can be improved

Are the conclusions supported by the results?

Can be improved

Are all figures and tables clear and well-presented?

Can be improved

3. Point-by-point response to Comments and Suggestions for Authors

General comment

Bacillus cereus, a foodborne pathogen known for causing diarrhea and vomiting, is increasingly resistant to drugs. This study identified a novel bacteriophage, Φ Bc24, in mud from Hanoi, Vietnam. Φ Bc24 exhibits specific targeting of Bacillus cereus, strong environmental stability, and safe genomic characteristics, indicating its significant potential as a biological control agent for food contamination.

The experimental design is reasonable, and the paper is logically structured. Supported by relevant data, the presentation is clear, and most conclusions are well-justified. However, there are some concerns regarding the experiment that need to be addressed before acceptance.

Response: We sincerely appreciate the time and effort the Reviewer have taken to evaluate our work. We are grateful for your insightful comments and suggestions, which have been invaluable in enhancing the quality of our manuscript.

Comments 1: Please check the whole manuscript carefully for avoiding grammar errors or spelling mistakes.

Response 1: The entire manuscript has been thoroughly revised to correct grammatical errors and spelling inconsistencies.

Comments 2: Please introduce the full name in the initial mention and link it with the corresponding acronym.(g. Line 23 ORFS, Line 125 PEG 800)

Response 2: We have made changes accordingly.

Comments 3: Materials and Methods: Please specify the primary instruments and reagents used, including their specific models and manufacturers.(e.g. LB, Centrifuge)

Response 3: This information has been added to the Materials and Methods section as suggested by the reviewer.

Comments 4: Line 90 to 97 The study employed four standard B. cereus strains, including VTCC 11273, and seven non-target bacteria, such as B. subtilis and E. coli. The test strains comprised species of the genus Bacillus and a few Gram-negative bacteria (Supplementary Table 1). Excluding common food industry contaminants, like close relatives of B. thuringiensis or B. anthracis, might result in an overestimation of host specificity.

Response 4: We sincerely thank the reviewer for this important suggestion. As suggested by the reviewer, we have included Bacillus thuringiensis in the host-range assay to better evaluate the specificity of phage ΦBc24 within the B. cereus group. The results showed that ΦBc24 was able to infect B. cereus strains but did not form plaques on B. thuringiensis strains tested, confirming its narrow host specificity.

Accordingly, we have updated the Materials and Methods section to describe the inclusion of B. thuringiensis in the bacterial panel and revised the Results section (in lines 255 - 261).

Comments 5: Line 162 to 171 The physicochemical stability experiments were conducted in SM buffer (pH, temperature, UV, etc.) without using food matrices. Buffers cannot simulate the effects of complex food components (such as fats, proteins) on phage activity. There is a lack of direct evidence to extrapolate the conclusions to 'food biocontrol agent' (Abstract, Line 27).

Response 5: We fully agree with the reviewer’s comment. The current stability assays were designed as a preliminary evaluation of the intrinsic robustness of ΦBc24 under different pH, temperature, and UV conditions. We have revised the Abstract and Discussion sections to clarify that these results represent preliminary data and do not directly confirm efficacy in food matrices.

Abstract (Line 27): The sentence “supporting its potential as a food biocontrol agent” has been revised to: “supporting its potential as a biocontrol candidate against the foodborne pathogen B. cereus.”

Discussion (in lines 590 - 593) to make it more clarify: “Additional studies are warranted to assess the stability and antibacterial activity of ΦBc24 in model food systems, which will help determine its practical feasibility as a biocontrol agent”.

Comments 6: Line 178 In vitro lytic activity assay experiment was observed for only 12 hours, and neither spore revival nor long-term antibacterial effects were evaluated, making it challenging to support its application in real food preservation scenarios. (Introduction, Line 44)

Response 6: We totally agree that the current lytic activity assay was limited to 12 hours and therefore did not evaluate long-term antibacterial effects or possible spore revival. As suggested by the reviewer, we have revised the Discussion section to clearly state this limitation. A new paragraph has been added in lines 445 - 449:

“The lytic activity of ΦBc24 was evaluated over 12 h to determine its short-term infection dynamics against B. cereus. Because Bacillus species can form spores that are resistant to phage infection, longer-term studies including 48–96 h growth monitoring and spore revival assays will be conducted in future work to fully assess the sustainability of ΦBc24’s antibacterial effect and its potential for food preservation applications.”

.

Comments 7: Figure 1 The scale lacks clarity. The enlarged image should include a scale to facilitate easier observation and comparison.

Response 7: We have added a clear scale bar to Figure 1 and improved image resolution to enhance visibility and comparison.   

Comments 8: Line 199 Why choose protein sequences of major capsid and large terminase subunits for phylogenetic analysis? Instead of analyzing the evolutionary branches of genes directly associated with lysis efficiency such as lysozyme and perforin. This may limit the understanding of the unique lysis mechanism of ΦBc24 and reduce its differential competitiveness compared to similar bacteriophages.

Response 8: We thank the reviewer for this valuable comment.

In this study, the sequences of the major capsid and large terminase subunits were chosen because they are well-established molecular markers for phage phylogenetic analysis (Wangchuk et al., 2021; de Jonge et al., 2024; Sada et al., 2024). The large terminase subunit encodes an ATP-driven motor responsible for genome packaging into the viral capsid and is highly conserved among Caudoviricetes phages, making it a reliable phylogenetic marker (Barylski et al., 2020; Koonin et al., 2020; de Jonge et al., 2024). Meanwhile, the major capsid protein (MCP) serves as the gold standard for lytic phage taxonomy (Sada et al., 2024; Rohwer & Edwards, 2020). The MCP gene encodes the principal structural protein forming the outer capsid that encapsulates and protects the viral genome during infection (Sada et al., 2024; Dewanggana et al., 2021). Due to its essential structural role and high conservation within specific phage families, the MCP gene is frequently used to construct phylogenetic trees and infer evolutionary relationships among phage isolates (Sada et al., 2024). Our results demonstrated that these two protein sequences serve as effective phylogenetic markers for Bacillus phages, comparable to using the whole-genome sequence.

We agree that genes involved in the phage lytic system (e.g., lysozyme, perforin, endolysins ) could provide additional insights as these components have adapted to target bacteria long before the evolution of multicellular organisms, and therefore they have been uniquely shaped by evolutionary pressure to affect bacterial cell walls (Ren et al., 2017; Vidová et al., 2014; Saint-Jean et al., 2025 ). However, the substantial diversity and the related gene copy number variation across phage lineages (Saint-Jean et al., 2025; Ren et al., 2017; Dewanggana et al., 2021; Vidová et al., 2014) may make them less reliable markers for reconstructing accurate phage phylogeny compared to more conserved structural or replication-associated genes, such as the major capsid or terminase subunits. We consider incorporating these lytic genes in future analyses to further investigate their evolutionary dynamics and unique lysis mechanisms.

Relevant information about the major capsid and large terminase subunits has been added to the Materials and Method (in lines 211 - 214) and Discussion sections (in lines 521 - 533).

Reference

Wangchuk, J., Chatterjee, A., Patil, S., Madugula, S. K., & Kondabagil, K. (2021). The coevolution of large and small terminases of bacteriophages is a result of purifying selection leading to phenotypic stabilization. Virology564, 13-25.

de Jonge, P. A., van den Born, B. J. H., Zwinderman, A. H., Nieuwdorp, M., Dutilh, B. E., & Herrema, H. (2024). Phylogeny and disease associations of a widespread and ancient intestinal bacteriophage lineage. Nature communications15(1), 6346.

Sada, T. S., Alemayehu, D. H., Tafese, K. M., & Tessema, T. S. (2024). Genomic analysis and characterization of lytic bacteriophages that target antimicrobial resistant Escherichia coli in Addis Ababa, Ethiopia. Heliyon10(22).

Barylski, J. et al. Analysis of spounaviruses as a case study for the overdue reclassification of tailed phages. Syst. Biol. 69, 110–123 (2020)

Koonin, E. V. & Yutin, N. The crAss-like phage group: How metagenomics reshaped the human virome. Trends Microbiol 28, 349–359 (2020)

Rohwer, F., & Edwards, R. (2002). The Phage Proteomic Tree: a genome-based taxonomy for phage. Journal of bacteriology184(16), 4529-4535.

Dewanggana, M. N., Waturangi, D. E., & Yogiara. (2021). Genomic characterization of bacteriophage BI-EHEC infecting strains of Enterohemorrhagic Escherichia coli. BMC Research Notes14(1), 459.

Ren, Q., Wang, C., Jin, M., Lan, J., Ye, T., Hui, K., ... & Han, G. Z. (2017). Co-option of bacteriophage lysozyme genes by bivalve genomes. Open Biology7(1), 160285.

Vidová, B., Šramková, Z., Tišáková, L., Oravkinová, M., & Godány, A. (2014). Bioinformatics analysis of bacteriophage and prophage endolysin domains. Biologia69(5), 541-556.

Saint-Jean, M., Claisse, O., Marrec, C. L., & Samot, J. (2025). Structural and genetic diversity of lysis modules in bacteriophages infecting the genus Streptococcus. Genes16(7), 842.

Comments 9: Despite bioinformatics predicting 'non-virulence,resistance genes,' experimental validation, such as hemolysis tests or detection of resistance gene expression, was not conducted. Relying solely on sequence prediction could overlook unannotated novel virulence factors, potentially posing biological safety risks, particularly when used as a food additive.

Response 9: We thank the reviewer for this insightful comment. As bacteriophages are obligate bacterial parasites with strict host specificity (de Jonge et al., 2024), direct assays such as hemolysis or resistance gene expression tests are not always applicable. Therefore, our analysis relied mainly on sequence-based predictions, though unannotated or inconsistently annotated genes in databases remain a limitation of this study. In the present work, gene annotation was mainly based on the PHROG database (Pharokka). Notably, the putative hemolysin gene (ORF0227) produced inconsistent results across annotation tools—BLASTn/p analysis, RAST annotations, and HHpred protein domain predictions suggested phage-related or membrane-associated functions rather than hemolytic activity.

A new sentence has been added in the Discussion in lines 509 - 512:

Although the current genomic analyses support the genetic safety of ΦBc24, we acknowledge the limitations of sequence-based inference, and additional biosafety evaluations via in vivo assessments, are required to confirm this finding.”

4. Response to Comments on the Quality of English Language

NA

5. Additional clarifications

 NA

We hope that the modifications adequately target all the concerns and drawbacks made by the Editor and Reviewer and that the paper is now accepted for publication in Current Issues in Molecular Biology. Once again, thank you so much for your consideration.

With my very best regards,

Dong Van Quyen, Assoc. Prof. Dr.

Reviewer 2 Report

Comments and Suggestions for Authors

I consider the topic highly interesting and with great potential. Nevertheless, the manuscript requires several improvements before publication, as outlined below:

Abstract

Some scientific names are not italicized.

Keywords

The first two keywords already appear in the title; I recommend replacing them.

Introduction

Lines 70–72: Reorganize and rewrite this section, as the phrasing is redundant.

Materials and Methods

  • Overall, the writing and structure of this section should be improved. In addition, a separate section describing the reagents is missing.
  • 2.2. Phage isolation and purification: The paragraph should be rewritten for clarity.
  • 2.3. Phage stock preparation: Please specify the medium used for all bacterial cultures, including Bacillus pumilus, Salmonella enterica, Vibrio parahaemolyticus, Escherichia coli, Staphylococcus aureus, and Lactobacillus kunkeei, as well as the procedures employed for their preparation.
  • Section 2.6. Multiplicity of infection: The beginning of this paragraph needs clearer wording.

Results

  • 3.3. Host range: Consider presenting these results in a more illustrative way.
  • Figures 2 and 4: Explain why only the results for B. cereus VTCC 11273 are shown. What about the other strains?

Discussion

  • The discussion needs to be strengthened. The authors mainly compare their results with previous studies but offer limited interpretation or explanation.
  • Lines 378–387: This paragraph is redundant; the information was already presented in the introduction.
  • Lines 388–398: Ensure all scientific names are italicized.
  • Lines 412–423: Provide a deeper discussion. In particular, link the phage characterization to explanations of its reported stabilities.

References

Review and adjust the references to align with the journal’s guidelines, especially for those from specialized journals.

Author Response

Genomic and Biological Insights of Bacteriophage ΦBc24 Targeting Bacillus cereus

Response to Reviewer 2 Comments

1. Summary

Thank you very much for taking the time to review this manuscript. Please find the detailed responses below and the corresponding revisions in track changes in the re-submitted files.

2. Questions for General Evaluation

Reviewer’s Evaluation

Does the introduction provide sufficient background and include all relevant references?

Yes

Is the research design appropriate?

Must be improved

Are the methods adequately described?

Must be improved

Are the results clearly presented?

Must be improved

Are the conclusions supported by the results?

Yes

Are all figures and tables clear and well-presented?

Must be improved

3. Point-by-point response to Comments and Suggestions for Authors

General comment

I consider the topic highly interesting and with great potential. Nevertheless, the manuscript requires several improvements before publication, as outlined below:

Comments 1: Abstract: Some scientific names are not italicized.

Response 1: We have made changes accordingly as the reviewer’s comment.

Comments 2: Keywords: The first two keywords already appear in the title; I recommend replacing them.

Response 2: The keywords have been revised as suggested by the reviewer. The final list is: Bastillevirinae; biocontrol; Caeruleovirus; food contamination; Herelleviridae; phage therapy

Comments 3: Materials and Methods

Overall, the writing and structure of this section should be improved. In addition, a separate section describing the reagents is missing.

•   2.2. Phage isolation and purification: The paragraph should be rewritten for clarity.

Response 3: We have revised this paragraph to make it more clarify.

Lines 112 - 117: “The presence of phages was determined by plaque formation on double-layer agar plates using the agar overlayspot assay method [39]. This assay was done by pouring top agar (LB consist of 0.7% agar) to the bottom agar (LB consist of 2% agar), where 150 μL of bacteriophage filtrate, and 150 μL of mid-log phase bacterial host culture were mixed in 5 mL of molten LB soft agar and poured onto LB agar plate, then followed by incubation at 37 °C overnight.”

•   2.3. Phage stock preparation: Please specify the medium used for all bacterial cultures, including Bacillus pumilus, Salmonella enterica, Vibrio parahaemolyticus, Escherichia coli, Staphylococcus aureus, and Lactobacillus kunkeei, as well as the procedures employed for their preparation.

We have made changes and added detail information as suggested by the reviewer. Lines 145 - 154: “Except for L. kunkeei, which was cultured in De Man–Rogosa–Sharpe (MRS) medium (Merck, Germany), all other bacterial strains were grown in LB medium with shaking at 150 rpm at 37 °C. For host range determination, 150 µl of each bacterial culture (OD₆₀₀ = 0.5–0.6) was mixed with 5 ml of molten soft agar (LB or MRS containing 0.7% agar) and overlaid onto a solid LB or MRS agar plate (2% agar). After the soft agar solidified for 15 min, 10 µl of the phage suspension was spotted onto the plate surface. The plates were incubated overnight at 37 °C, and the resulting lysis zones were examined. Phage host range was assessed based on plaque clarity: “++” indicated clear plaques; “+” indicated faint plaques; and “–” indicated no plaque formation.”

•   Section 2.6. Multiplicity of infection: The beginning of this paragraph needs clearer wording.

Multiplicity of infection has been revised to: The optimal multiplicity of infection (MOI)”

Comments 4: Results

•  3.3. Host range: Consider presenting these results in a more illustrative way.

Response 4: We thank the reviewer for this comment.

Section 3.3. Host range has been revised accordingly as the reviewer’s suggestion

•     Figures 2 and 4: Explain why only the results for B. cereus VTCC 11273 are shown. What about the other strains

As clarified in Section 3.1, only B. cereus VTCC 11273 is shown in Figures 2 and 4 because it exhibited the highest lytic activity and was therefore selected as the propagation host. Presenting the results for B. cereus VTCC 11273 allows for a more focused and representative analysis.

Comments 5: Discussion

•   The discussion needs to be strengthened. The authors mainly compare their results with previous studies but offer limited interpretation or explanation.

Response 5: The Discussion has been strengthened by providing deeper interpretation of the results and linking them to potential applications and biological implications, beyond comparisons with previous studies.

•   Lines 378–387: This paragraph is redundant; the information was already presented in the introduction.

This paragraph has been revised accordingly as suggested by the reviewer.

•   Lines 388–398: Ensure all scientific names are italicized.

All scientific names have been carefully checked and italicized as appropriate.

•   Lines 412–423: Provide a deeper discussion. In particular, link the phage characterization to explanations of its reported stabilities.

This section has been revised to provide a deeper discussion, linking the observed stability characteristics of phage ΦBc24 to its structural features and potential ecological adaptations, thereby highlighting its robustness and practical relevance as a biocontrol agent.

Comments 6: Review and adjust the references to align with the journal’s guidelines, especially for those from specialized journals.

Response 6: This has been revised accordingly as the reviewer’s comment.

4. Response to Comments on the Quality of English Language

Point 1: The English could be improved to more clearly express the research.

Response 1: We sincerely thank the reviewer for this suggestion. The entire manuscript has been carefully revised for English grammar, clarity, and academic style with the assistance of a native English-speaking colleague experienced in scientific writing. We believe the language of the revised version now clearly and accurately conveys the research content.

5. Additional clarifications

 NA

We hope that the modifications adequately target all the concerns and drawbacks made by the Editor and Reviewer and that the paper is now accepted for publication in Current Issues in Molecular Biology. Once again, thank you so much for your consideration.

With my very best regards,

Dong Van Quyen, Assoc. Prof. Dr.

Reviewer 3 Report

Comments and Suggestions for Authors

The aims of the study should be clearly stated in the abstract.

How can phages be used towards food safety? What is missing before they can be commercially available? What are the legal constraints? And the advantages? Please critically discuss this topic.

What will be this phage's true impact and "potential as a biocontrol agent to mitigate B. cereus contamination in food production"? Are we close to implementing this control strategy in the food industry? In Europe? In the USA? In China? Worldwide?

Author Response

Genomic and Biological Insights of Bacteriophage ΦBc24 Targeting Bacillus cereus

Response to Reviewer 3 Comments

1. Summary

Thank you very much for taking the time to review this manuscript. Please find the detailed responses below and the corresponding revisions in track changes in the re-submitted files.

2. Questions for General Evaluation

Reviewer’s Evaluation

Does the introduction provide sufficient background and include all relevant references?

Can be improved

Is the research design appropriate?

Yes

Are the methods adequately described?

Yes

Are the results clearly presented?

Yes

Are the conclusions supported by the results?

Can be improved

Are all figures and tables clear and well-presented?

Yes

3. Point-by-point response to Comments and Suggestions for Authors

Comments 1: The aims of the study should be clearly stated in the abstract.

Response 1: The abstract has been revised to clearly state the aims of the study as suggested by the reviewer.

Comments 2: How can phages be used towards food safety? What is missing before they can be commercially available? What are the legal constraints? And the advantages? Please critically discuss this topic.

Response 2: We thank the reviewer for this valuable suggestion. We have accordingly included the relevant information in the revised manuscript (lines 394 – 407, 594 – 603, 603 – 610 and 611 - 618) provide a concise overview of the application potential, limitations, and regulatory context of bacteriophages in food safety:

“Phages are natural antibacterial agents with strong biocontrol potential against major and emerging foodborne pathogens such as Bacillus, Campylobacter, E. coli, Listeria monocytogenes, Salmonella, Shigella, and Vibrio spp. (Endersen & Coffey, 2020; Bumunang et al., 2023; Goodridge & Bisha, 201; Narayanan et al., 2024). During the lytic cycle, each phage produces numerous progeny that rapidly lyse host cells, effectively suppressing pathogens (Wang & Zhao, 2022). Compared with traditional antimicrobials, phages are highly specific, preserve beneficial microflora, have no adverse effects on humans, do not alter food quality, are easy and low-cost to produce, stable under diverse conditions, and self-replicating—eliminating repeated dosing (Brovko et al., 2012; Singh, 2018). These attributes make them valuable for both detection and control of pathogens throughout the food chain, including post-harvest surface decontamination, antimicrobial packaging, equipment sanitation, animal pre-harvest treatment, and phage-based diagnostics (Brovko et al., 2012; Singh, 2018; Bumunang et al., 2023).

These phages are increasingly recognized as safe (GRAS) for food use and considered organic and legitimate (Narayanan et al., 2024; Amjad et al., 2024; Dhulipalla et al., 2025). Since the first FDA-approved phage product (ListShield™, 2006), multiple preparations—including SalmoFresh™ (2013), ShigaShield™ (2017), and Salmonelex™ (2013)—have entered the market from companies such as Intralytix, Micreos, FINK TEC, Passport Food Safety Solutions, and Phagelux (Endersen & Coffey, 2020; Wang & Zhao, 2022). These developments highlight growing confidence in phage efficacy and safety. However, widespread commercialization is constrained by factors such as narrow host range, inconsistent database annotations and numerous uncharacterized ORFs complicating safety interpretation, formulation stability, limited real-world validation (in large-scale and multi-site trials) and evaluation standards (Brovko et al., 2012; Wang & Zhao, 2022; Amjad et al., 2024; Dhulipalla et al., 2025).

Legal constraints and regulatory frameworks vary internationally—while GRAS/FCN approvals exist in the U.S., phages are classified as novel foods in the EU and require EFSA evaluation (Imran et al., 2023; EFSA, 2024). Canada, Australia, Switzerland, Israel, and New Zealand have also approved phage products, often aligning with U.S. precedents. However, slow EU regulatory adoption continues to delay implementation. Therefore, harmonized evaluation standards and transparent risk–benefit communication remain essential for global (Dhulipalla et al., 2025).

To ensure safe and responsible commercialization, standardized genomic screening pipelines, harmonized annotation practices; and large-scale and multi-site field trials are needed to confirm robustness and monitor resistance. Improved formulation technologies can enhance stability and delivery, while early regulatory engagement may streamline approval. Lastly, transparent, evidence-based consumer communication is vital to build public understanding and trust in phage-based solutions (Brovko et al., 2012; Bumunang et al., 2023; Dhulipalla et al., 2025).

Reference

Endersen, L.; Coffey, A. The use of bacteriophages for food safety. Current Opinion in Food Science 2020, 36, 1–8, doi:https://doi.org/10.1016/j.cofs.2020.10.006.

Bumunang, E.W.; Zaheer, R.; Niu, D.; Narvaez-Bravo, C.; Alexander, T.; McAllister, T.A.; Stanford, K. Bacteriophages for the Targeted Control of Foodborne Pathogens. Foods 202312, 2734. https://doi.org/10.3390/foods12142734

Goodridge, L. D., & Bisha, B. (2011). Phage-based biocontrol strategies to reduce foodborne pathogens in foods. Bacteriophage1(3), 130–137. https://doi.org/10.4161/bact.1.3.17629

Zhihui Wang, Xihong Zhao, The application and research progress of bacteriophages in food safety, Journal of Applied Microbiology, Volume 133, Issue 4, 1 October 2022, Pages 2137–2147, https://doi.org/10.1111/jam.15555

Brovko, L. Y., Anany, H., & Griffiths, M. W. (2012). Chapter Six - Bacteriophages for Detection and Control of Bacterial Pathogens in Food and Food-Processing Environment. In J. Henry (Ed.), Advances in Food and Nutrition Research (Vol. 67, pp. 241–288). Academic Press. https://doi.org/https://doi.org/10.1016/B978-0-12-394598-3.00006-X

Singh, V. (2018). Bacteriophage-mediated biosensors for detection of foodborne pathogens. In Microbial bioprospecting for sustainable development (pp. 353-384). Singapore: Springer Singapore.

Narayanan, K. B., Bhaskar, R., & Han, S. S. (2024). Bacteriophages: Natural antimicrobial bioadditives for food preservation in active packaging. International Journal of Biological Macromolecules, 276, 133945. https://doi.org/https://doi.org/10.1016/j.ijbiomac.2024.133945

Amjad, N., Naseer, M. S., Imran, A., Menon, S. V., Sharma, A., Islam, F., … Shah, M. A. (2024). A mini-review on the role of bacteriophages in food safety. CyTA - Journal of Food22(1). https://doi.org/10.1080/19476337.2024.2357192

Dhulipalla, H., Basavegowda, N., Haldar, D. et al. Integrating phage biocontrol in food production: industrial implications and regulatory overview. Discov Appl Sci 7, 314 (2025). https://doi.org/10.1007/s42452-025-06754-3

Hyla, K.; Dusza, I.; Skaradzińska, A. Recent Advances in the Application of Bacteriophages against Common Foodborne Pathogens. Antibiotics 2022, 11, 1536. https://doi.org/10.3390/antibiotics11111536

Comments 3: What will be this phage's true impact and "potential as a biocontrol agent to mitigate B. cereus contamination in food production"? Are we close to implementing this control strategy in the food industry? In Europe? In the USA? In China? Worldwide?

Response 3: We appropriate the reviewer’s comment. The relevant information has been accordingly added in the MS in lines 562 – 567 and 558 – 561.

“Our results demonstrated the potential of phage ΦBc24 as a biocontrol candidate against the foodborne pathogen B. cereus in vitro, contributing additional evidence, resources, and foundational knowledge regarding B. cereus phages and their potential applications in food production since the number of available B. cereus-infecting phage isolates are relatively small and products specifically targeting B.cereus remain limited (Seol et al., 2024).

Although phages targeting B. cereus have gained growing attention for their potential to reduce B. cereus contamination in food production across several countries, including Indonesia (2024) (Rizkinata et al., 2024), China (2024) (Huang et al., 2024; Chen et al., 2024), Korea (2017 and 2021) (Oh et al., 2017; Shin et al., 2011) Belgium (2014) (Gillis et al., 2014), large-scale implementation is still in its early stages. The ongoing advances in formulation, genomic characterization, and regulatory harmonization suggest that phage-based control strategies are approaching practical feasibility worldwide (Dhulipalla et al., 2025).”

References

Seol, H., Kim, B. S., & Kim, M. (2024). Control of Bacillus cereus in rice cake and food contact surfaces with novel Becedseptimavirus genus phage BCC348 and its partial SPOR domain-containing endolysin LysBCC348. LWT, 213, 117034. https://doi.org/https://doi.org/10.1016/j.lwt.2024.117034

Rizkinata, D., Kusnadi, V.C., Waturangi, D.E. et al. Isolation and molecular characterization of bacteriophages isolated from lake water and their application in foods against Bacillus cereusBMC Res Notes 18, 364 (2025). https://doi.org/10.1186/s13104-025-07436-4

Huang, Z., Yuan, X., Zhu, Z., Feng, Y., Li, N., Yu, S., Li, C., Chen, B., Wu, S., Gu, Q., Zhang, J., Wang, J., Wu, Q., & Ding, Y. (2024). Isolation and characterization of Bacillus cereus bacteriophage DZ1 and its application in foods. Food Chemistry, 431, 137128. https://doi.org/https://doi.org/10.1016/j.foodchem.2023.137128

Chen, B. O., Huang, Z., Yuan, X., Li, C., Wang, J., Chen, M., Xue, L., Zhang, J., Wu, Q., & Ding, Y. U. (2024). Isolation and characterization of two phages against emetic Bacillus cereus and their potential applications. Food Frontiers, 5, 2305–2318. https://doi.org/10.1002/fft2.425

Oh, H., Seo, D.J., Jeon, S.B. et al. Isolation and Characterization of Bacillus cereus Bacteriophages from Foods and Soil. Food Environ Virol 9, 260–269 (2017). https://doi.org/10.1007/s12560-017-9284-6

Shin, H., Bandara, N., Shin, E., Ryu, S., & Kim, K.-p. (2011). Prevalence of Bacillus cereus bacteriophages in fermented foods and characterization of phage JBP901. Research in Microbiology, 162(8), 791–797. https://doi.org/https://doi.org/10.1016/j.resmic.2011.07.001

Gillis, A.; Mahillon, J. Phages Preying on Bacillus anthracisBacillus cereus, and Bacillus thuringiensis: Past, Present and Future. Viruses 20146, 2623-2672. https://doi.org/10.3390/v6072623

4. Response to Comments on the Quality of English Language

NA

5. Additional clarifications

 NA

We hope that the modifications adequately target all the concerns and drawbacks made by the Editor and Reviewer and that the paper is now accepted for publication in Current Issues in Molecular Biology. Once again, thank you so much for your consideration.

With my very best regards,

Dong Van Quyen, Assoc. Prof. Dr.

Round 2

Reviewer 2 Report

Comments and Suggestions for Authors

The authors have addressed all the suggested corrections, and the inclusion of new graphics has provided greater clarity to the research results.

Author Response

Genomic and Biological Insights of Bacteriophage ΦBc24 Targeting Bacillus cereus

Response to Reviewer 2 Comments

1. Summary

Thank you very much for taking the time to review this manuscript. Please find the detailed responses below and the corresponding revisions in track changes in the re-submitted files.

2. Questions for General Evaluation

Reviewer’s Evaluation

Does the introduction provide sufficient background and include all relevant references?

Yes

Is the research design appropriate?

Yes

Are the methods adequately described?

Yes

Are the results clearly presented?

Yes

Are the conclusions supported by the results?

Yes

Are all figures and tables clear and well-presented?

Yes

Comments and Suggestions for Authors

The authors have addressed all the suggested corrections, and the inclusion of new graphics has provided greater clarity to the research results.

Response: We sincerely appreciate the time and effort the Reviewer have taken to evaluate our work. We are grateful for your insightful comments and suggestions, which have been invaluable in enhancing the quality of our manuscript.

We hope that the paper is now accepted for publication in Current Issues in Molecular Biology. Once again, thank you so much for your consideration.

With my very best regards,

Dong Van Quyen, Assoc. Prof. Dr.

Reviewer 3 Report

Comments and Suggestions for Authors

The authors have significantly improved their manuscript follwing the sugestions made to the original version. I suggest that the term "microflora" be replaced by "microbiota".

Author Response

Genomic and Biological Insights of Bacteriophage ΦBc24 Targeting Bacillus cereus

Response to Reviewer 3 Comments

1. Summary

Thank you very much for taking the time to review this manuscript. Please find the detailed responses below and the corresponding revisions in track changes in the re-submitted files.

2. Questions for General Evaluation

Reviewer’s Evaluation

Does the introduction provide sufficient background and include all relevant references?

Can be improved

Is the research design appropriate?

Yes

Are the methods adequately described?

Yes

Are the results clearly presented?

Yes

Are the conclusions supported by the results?

Can be improved

Are all figures and tables clear and well-presented?

Yes

3. Comments and Suggestions for Authors

Comment 1. The authors have significantly improved their manuscript follwing the sugestions made to the original version. I suggest that the term "microflora" be replaced by "microbiota".

Response 1: We sincerely appreciate the time and effort the Reviewer have taken to evaluate our work. We are grateful for your insightful comments and suggestions, which have been invaluable in enhancing the quality of our manuscript.

The term "microflora" has been replaced by "microbiota" as suggested by the reviewer in lines 406 and 429.

Comment 2: The introduction can be improved.

Response 2: We sincerely thank the reviewer for this important comment. A new paragraph has been added to the introduction in lines 58-66 and this section has also been revised accordingly.

In Vietnam, B. cereus has also been associated with several foodborne poisoning incidents in recent years. Between 2020 and 2025, multiple outbreaks were linked to contamination by B. cereus, either alone or in combination with Escherichia coli and Salmonella. Notably, an outbreak in Da Nang province in 2020 affected 230 individuals, while another at a school in Nha Trang province in 2022 led to the hospitalization of more than 600 students (National Institute for Food Control, Vietnam, 2024). More recently, a food poisoning event in Phu Tho province in 2025 affected 438 workers (Ministry of Health, 2025). These incidents highlight the persistent public health risk posed by B. cereus contamination in food products.”

Comment 2: The conclusions supported by the results can be improved.

Response 2: We thank the reviewer for this insightful comment. The conclusions have been revised as suggested by the reviewer.

“This study reports, for the first time, the isolation and genomic characterization of the B. cereus phage ΦBc24 in Vietnam. Phage ΦBc24 represents a novel lytic member of the family Herelleviridae, subfamily Bastillervirinae, and genus Caeruleovirus. Detailed biological and genomic analyses demonstrated its pronounced host specificity, potent lytic activity, and absence of undesirable genetic elements, underscoring its potential as a safe and effective biocontrol agent against B. cereus. Collectively, these findings reinforce the promise of bacteriophage-based biocontrol as a sustainable and targeted alternative to conventional antimicrobial approaches for mitigating foodborne pathogens and enhancing food safety. Further investigations are warranted to optimize formulation and delivery strategies, evaluate long-term stability and safety, and address potential implications for human health and regulatory acceptance.”

4. Response to Comments on the Quality of English Language

NA

5. Additional clarifications

 NA

We hope that the modifications adequately target all the concerns and drawbacks made by the Editor and Reviewer and that the paper is now accepted for publication in Current Issues in Molecular Biology. Once again, thank you so much for your consideration.

With my very best regards,

Dong Van Quyen, Assoc. Prof. Dr.